# Relationship Between the Total Quality Recovery Scale and Race Performance in Competitive College Swimmers over Two Seasons

**DOI:** 10.3390/sports13050139

**Published:** 2025-04-30

**Authors:** Tsuyoshi Kato, Ryota Kasugai, Kensuke Sakai

**Affiliations:** 1Sports Promotion Center, Tokai University, Hiratsuka 259-1292, Japan; kt6597@tokai.ac.jp (T.K.); kasugai.ryota.n@tokai.ac.jp (R.K.); 2Faculty of Pharmaceutical Sciences, Josai International University, Togane 283-8555, Japan

**Keywords:** swimming performance, recovery state, total quality recovery, daily monitoring, tapering strategy, subjective recovery assessment, training load management

## Abstract

*Background:* Tapering, a period of reduced training load following intense training, contributes to performance enhancement. However, research on recovery status during tapering is limited. This study investigates the impact of recovery status on race performance. *Methods:* Total quality recovery (TQR) scale scores were monitored over two seasons in 22 college competitive swimmers (age: 19.7 ± 1.8 years), including 6 females. They participated voluntarily in the study. Rolling averages (TQR*ra*) and exponentially weighted moving averages (TQR*ewma*) over 7, 14, 21, and 28 days before the race were calculated. Performance data from 550 race days were analyzed by quartiles, and odds ratios were computed for TQR-related variables against race performance improvements. *Results:* The median TQR was 11 (interquartile range: 10–13). Seasonal bests were achieved in 31.6% of races (174 of 550). The highest odds ratios (ORs) for performance improvement in the highest quartile (Q4) of TQR and z-TQR were 3.13 (*p* < 0.001) and 4.35 (*p* < 0.001), respectively. Significant ratios for TQR*ewma* were observed for 7d:21d (OR: 2.62, *p* < 0.001) and 7d:28d (OR: 2.48, *p* < 0.001) comparisons. *Conclusions:* Better recovery status on race day has been associated with improved swimming performance. Additionally, optimizing the TQR*ewma* ratio of the most recent 7 days compared to the preceding 21 to 28 days may further enhance race performance. It highlights the need to monitor an athlete’s recovery over several weeks as an important pre-race strategy.

## 1. Introduction

Achieving optimal performance in competitive swimming necessitates enhancements across physical, technical, physiological, and psychological dimensions [1,2]. Swimming events range widely from 50 m sprints to 1500 m endurance races. Many swimmers strive to improve their performance by increasing maximum strength and aerobic endurance [3]. However, some swimmers have struggled to improve their personal records for years.

To optimize performance, a tapering phase is commonly implemented before key races [4,5,6]. Tapering refers to the recovery period before a competition, during which the training load (TL) is reduced to alleviate stress and maximize race-day performance [7,8,9,10]. Tapering methods are discussed in terms of training volume, intensity, frequency, and duration before the race. Bosquet et al.’s meta-analysis revealed that effective tapering strategies often involve a gradual reduction in training volume by 41–60% over 8–14 days while maintaining training intensity and frequency [11]. They further emphasized the importance of maintaining training intensity and frequency during tapering. Aouani et al. found that a 2-week tapering period improved performance in the 50 m front crawl test in competitive swimmers [12]. Mujika and Padilla noted that a non-linear reduction in training volume can alleviate the physiological and psychological stress caused by daily training and optimize performance [9].

Research on endurance [13] and strength athletes [14] supports the effectiveness of tapering for enhancing performance. Additionally, performance improvements can result from high-intensity or high-volume training preceding the tapering phase [13,15]. Training involves two aspects: fitness enhancement through chronic load and fatigue accumulation due to acute load. Combining a period of high chronic load with a sudden reduction in load can lead to improved performance [16,17].

Performance improvements can be expected by facilitating recovery while preserving the benefits of high-intensity training. However, research on tapering has primarily focused on TL, such as volume and intensity. Conversely, the impact of TL on performance during high-intensity training and subsequent tapering, which are planned by coaches using external and internal TLs, has been investigated [18,19]. Aubry et al. evaluated the performance of triathletes during a 3-week overload training period followed by a 4-week tapering period. Those who adapted to overload training showed significant performance improvements in the 2nd week of the tapering phase compared to those who did not. Additionally, some participants could not adapt to overload training and experienced functional overreach [20]. This finding underscores the importance of monitoring both the TL imposed during high-intensity training and the athlete’s recovery from fatigue.

Total quality recovery (TQR) is recognized as a scale for assessing recovery status [21]. Although various physiological and psychological indicators have been proposed for fatigue-recovery measures, Saw et al. suggested that subjective indicators tend to be more sensitive than objective ones [22], recommending the use of brief scales for daily monitoring [23]. TQR, a single-item scale related to recovery status, imposes a minimal burden on participants, thereby ensuring high compliance. Studies have reported using TQR as a recovery scale among soccer [24,25], volleyball [26,27,28], and basketball players [29]. Most of these studies investigated the relationship between internal TL (Session Rating of Perceived Exertion [sRPE]) and injury occurrence; however, research on swimmers is limited. Crowcroft et al. reported that the 7-day rolling average of TQR (TQR*ra*) impacts race performance in competitive swimmers [30], suggesting that sufficient recovery before a race positively affects performance. However, the effect of overload training on recovery was not mentioned. Continuation of overload training before the tapering period is expected to delay recovery from fatigue. Additionally, Collette et al. reported a time lag and individual variability between sRPE as an internal TL and subjective recovery [31]. Therefore, training strategies before major races should focus on external or internal TL, along with monitoring the athlete’s recovery status.

In this study, we investigated the impact of recovery status on race performance. We monitored the TQR score longitudinally in competitive swimmers throughout the season and examined their pre-race recovery status using TQR-related variables. The findings from this study will contribute to understanding recovery status conducive to optimal performance and assist in planning and adjusting training programs from the overload period to the tapering phase before important races.

## 2. Materials and Methods

### 2.1. Participants

The study monitored TQR, a subjective recovery scale, daily for 36 competitive swimmers from a single university swim team over two seasons (99 weeks) between October 2020 and August 2022. Participants were recruited during team meetings, and all swimmers volunteered. The study focused on 22 swimmers who had been members of the team for at least one season during the monitoring period and had a submission rate of TQR score above 85% (missing no more than 1 day per week) (Figure 1). Some swimmers participate in approximately 10 races in a season [32].

Among the 22 participants, 6 were female, with an average age of 19.7 ± 1.8 years, average competitive experience of 13.9 ± 3.7 years, and average height of 169.3 ± 7.8 cm. Their average body weight, measured using bioelectrical impedance analysis (InBody 470, TANITA, Tokyo, Japan), was 63.9 ± 9.4 kg; the average lean body mass was 53.7 ± 8.4 kg, and the average body fat mass was 10.2 ± 2.6 kg. Participants included seven swimmers specializing in freestyle, three in backstroke, six in breaststroke, three in butterfly, and three in individual medley. The average competitive level, according to the FINA points classification by Ruiz-Navarro et al., was Level 3 (739 ± 49), with one swimmer each at Level 2 and Level 4 [33]. This study was approved by the Josai International University Research Ethics Committee (Approval Number: 15W200012) and conducted in accordance with the Declaration of Helsinki. The research protocol was explained orally to the participants, and written consent was obtained.

### 2.2. Training and Race Schedule

During the monitoring period, participants typically engaged in 10 to 11 swim training sessions and two to three strength training sessions per week. After 30 min of stretching on land, each swim session lasted 2 h, including a 10 min warm-up and a 15 min cool-down. The training programs were designed by two coaches, with four different programs tailored to the participants’ specific events and distances. The average weekly swimming distance during the monitoring period was 42,400 ± 16,324 m. Strength training sessions were conducted under the guidance of an expert [34] and lasted 60 to 90 min per session.

Team training was suspended during weeks 8 and 9 of the 2020/2021 season owing to an infectious disease; week 53 was considered an off-season between the two seasons.

In the 2020/2021 season (weeks 1 to 52), the participants engaged in 7 short-course and 15 long-course events. Conversely, in the 2021/2022 season (weeks 54 to 99), they participated in 10 short-course and 15 long-course events. The events varied in duration, with some being held over a single day and others spanning multiple days. Participants selected races based on personal preference and did not enter all events. Although some swimmers tapered prior to events, many participated without adequate adjustment. All race times were recorded electronically.

### 2.3. TQR Scores and TQR-Related Variables

Participants provided daily TQR scores to assess their recovery status. The TQR scale is divided into 15 points ranging from 6 (very poor recovery) to 20 (very good recovery), with 13 being the baseline [21]. Specifically, participants received an email each morning at 5 a.m., prompting them to fill out a TQR scale created online using Google Forms. Since morning training began at 6 a.m., they were required to input the TQR scores between waking up and the start of training. To calculate rolling averages (TQR*ra*) and exponentially smoothed moving averages (TQR*ewma*) over a specific period, they were also encouraged to respond on holidays and days without training. TQR scores obtained for each participant were used to calculate TQR*ra* and TQR*ewma* for the previous 7, 14, 21, and 28 days. The TQR*ewma* was calculated according to Williams S. et al. as follows [35]:TQR*ewma* today = TQR today × λa + ((1 − λa) × TQR*ewma* yesterday),
where λa = 2/(N + 1).

For calculating the TQR*ewma* on the first day of observation, a TQR score of 13 was assumed for the day before the observation period. Missing data were filled in using the Last Observation Carried Forward (LOCF) method. Subsequently, six ratios were calculated using the TQRra and TQR*ewma* (7d:14d, 7d:21d, 7d:28d, 14d:21d, 14d:28d, and 21d:28d). Additionally, these TQR ratios were converted to z-scores for each participant, allowing for analysis of intra-individual variability. The z-score is a standardized score that follows a normal distribution with a mean of 0 and a standard deviation of 1. Each individual’s z-score was calculated by dividing the difference between their daily TQR score and their average TQR score during the observation period by the standard deviation.

### 2.4. Performance

Race results for each participant were compared to their season-best times, with performance percentages calculated accordingly. The best time from the previous season was used as the benchmark for the first race of the current season. Only results from participants’ specialized events were considered, and if multiple races occurred on the same day, only the best performance was included.

### 2.5. Statistical Analysis

A total of 11,429 days of TQR data were collected during the observation period. The relationship between the weekly average of TQR score and swimming distance was evaluated using Spearman’s rank correlation coefficient, with the following thresholds: trivial (<0.1), small (0.1–0.3), moderate (0.3–0.5), large (0.5–0.7), very large (0.7–0.9), and nearly perfect (>0.9). Among the 11,429 days of data, 550 days of race data were extracted for further analysis. These TQR-related variables (TQR, TQR ratio, and each z-score) were divided into quartiles; a binary logistic regression analysis was performed to examine race performance (not improved = 0; improved = 1) as the dependent variable, using the lowest quartile (Q1) as the reference. Additionally, a trend analysis of the quartiles and race performance was conducted using the Cochran–Armitage test. Furthermore, the ratio of race time to season-best time for each quartile was analyzed using the Kruskal–Wallis test. Post hoc comparisons between groups were made with the Mann–Whitney U test, applying Bonferroni adjustments, as the Shapiro–Wilk test indicated non-normal distribution. Statistical procedures were performed using IBM SPSS Statistics (version 26, IBM Japan, Ltd., Tokyo, Japan), with a significance level set at less than 5%. Results are presented as medians and interquartile ranges (IQRs).

## 3. Results

### 3.1. Training Volume and TQR Distribution

Figure 2 shows the weekly averages of TQR scores and swimming distance throughout the observation period. A significant negative correlation was found between these variables (ρ = −0.601, *p* < 0.001). The average compliance rate for reporting TQR scores among participants was 97.2 ± 2.2%. Table 1 shows the distribution of TQR scores during the observation period (*n* = 11,429) and on race days (*n* = 550). The mode for both distributions was 13, with a median (IQR) of 11 (10–13) throughout the observation period and 12 (10–13) on race days. TQR scores below 12, indicating insufficient recovery, were observed on 65.9% of the days in the observation period and 55.1% of race days.

The weekly average TQR scores (A) and weekly average swim distances (B) during the observational period (99 weeks) are shown (*n* = 11,429). All data are expressed as Mean ± SD.

### 3.2. TQR-Related Variables and Race Performance

Seasonal-best records were achieved on 174 (31.6%) of 550 race days. Table 2 shows the impact of each variable, divided into quartiles, on race performance. Higher quartiles of TQR on race days (Q3 and Q4) demonstrated higher odds ratios for achieving seasonal bests, at 2.22 (95% CI: 1.28–3.87) and 3.13 (95% CI: 1.84–5.30), respectively. The Cochran–Armitage trend test also revealed a significantly improved race performance with higher TQR quartiles (*p* < 0.001).

For TQR ratios calculated using rolling averages on race days (TQR*ra*), significant differences were observed in the 7d:14d and 7d:28d ratios, as indicated by the Cochran-Armitage trend test. The odds ratios for the highest quartile (Q4) were 1.68 (95% CI: 1.01–2.80) for the 7d:21d ratio and 1.72 (95% CI: 1.03–2.87) for the 7d:28d ratio. Conversely, TQR ratios using TQR*ewma* showed odds ratios above 2 in Q4, with the 7d:21d and 7d:28d ratios yielding odds ratios of 2.62 (95% CI: 1.56–4.41, *p* < 0.001) and 2.48 (95% CI: 1.47–4.19, *p* < 0.001), respectively. Moreover, significant differences were observed in all variables, as indicated by the Cochran–Armitage trend test.

Given that TQR scores are subjective measures, individual variability was considered. Table 3 shows the odds ratios for z-TQR ratios related to race performance. Similarly to TQR, the z-TQR in Q4 displayed an odds ratio of 4.35 (95% CI: 2.56–7.41, *p* < 0.001) on race days. Among the z-TQR ratios, z-TQR*ewma* demonstrated higher odds ratios compared to z-TQR*ra*, with the highest odds ratio observed in z-TQR*ewma* 7d:21d at 2.48 (95% CI: 1.47–4.18, *p* < 0.001), comparable to TQR*ewma*.

Assessing the effect of quartiles as independent variables on race performance using the Kruskal–Wallis test revealed that all variables showed significant differences except for the 21d:28d ratio in TQR*ra* and z-TQR*ra* and the 14d:28d ratio in z-TQR*ra*. Q4 showed significantly lower values than Q1. The lowest median values were observed for TQR*ewma* and z-TQR*ewma* in the 7d:21d ratio at 100.2% (Table 4 and Table 5).

## 4. Discussion

This study examined the impact of recovery status over the 4 weeks leading up to race day on the performance of collegiate competitive swimmers, using the TQR across two seasons. To our knowledge, limited research has examined the effect of daily monitoring of TQR scores on race performance [30]. Stone et al. noted that although many studies assessed psychological states during the tapering period—primarily using POMS (Profile of Mood States [18%]) and RPE (Rating of Perceived Exertion [14%])—only a small fraction (9%) examined recovery-stress status, with no studies including TQR. Most studies assessed subjective measures during specific periods with different training objectives. However, no studies monitored these measures daily over a long period [36].

Race performance is influenced by physical, technical, physiological, and psychological factors, with adequate recovery being critical [37]. Our findings indicated that higher TQR scores on race days were associated with improved performance; specifically, the odds ratios for Q4 were 3.13 for TQR and 4.35 for z-TQR compared to those for Q1. These findings suggest that enhanced recovery status on race day improves race performance. Furthermore, increasing the ratio of TQR*ewma* in the week before the race compared to earlier periods showed significant odds ratios in Q4 (2.62 for 7d:21d and 2.48 for 7d:28d), supporting the notion that proper tapering after high-intensity training is crucial. After prolonged high-intensity training, recovery status may not reach optimal levels for several days. Howle et al. highlighted that continuously imposing high internal loads can accumulate fatigue and delay recovery [38]. Additionally, a time lag (0–6 days) exists between internal load and recovery, with significant individual variations reported [31]. Bosquet et al. reported that an 8–14 day tapering period is suitable for swimmers [11]. Additionally, a review by Stone et al. indicates that most studies have employed tapering periods ranging from 7 to 14 days [36], aligning with our findings.

Furthermore, training strategies before significant races often focus on external loads planned by coaches, which may not align with the athletes’ internal loads and recovery statuses. A meta-analysis by Inoue et al. found no significant difference between athletes’ perceived internal loads (sRPE) during training and how coaches perceived those loads. However, when TL is lower, coaches may underestimate the sRPE experienced by athletes [39]. This suggests that during the tapering phase, coaches might not fully understand the fatigue status of athletes. A report on basketball players showed that coaches underestimated players’ recovery status using TQR scores [40]. Training planned without an accurate understanding of the recovery status of athletes during the tapering period may lead to further recovery delays. Therefore, the importance of continuously monitoring recovery status is highlighted.

The TQR ratio calculated using rolling averages (RA) and exponentially weighted moving averages (EWMA) in this study aligns with the Acute Chronic Workload Ratio (ACWR), a measure used to assess internal load [41]. ACWR helps detect injury risks associated with sudden changes in internal load, with a preference for EWMA over RA [35,42]. In our study, TQR ratios using EWMA (TQR*ewma*) demonstrated higher odds ratios than those using RA (TQR*ra*), aligning with the Banister’s Fitness-Fatigue model, which suggests that fatigue recovery follows an exponential pattern [16,17]. However, a significant limitation of using EWMA is the impact of missing data, which can influence the day’s EWMA value and subsequent values. Therefore, many studies that use EWMA often rely on weekly averages, potentially excluding daily missing data. Relying on weekly averages rather than daily data may not accurately reflect a recovery from accumulated fatigue on race day. In this study, participants maintained high compliance in reporting their TQR scores during the observation period (97.2 ± 2.2%), with missing values addressed using the last observation carried forward (LOCF) method [43]. If a race occurs mid-week after a tapering phase, substantial differences in recovery status could be observed in the days leading up to the race as well as after the race. Therefore, we adopted LOCF to accurately reflect the recovery status from the previous day. Alternative imputation methods, such as regression models, have been suggested for addressing missing data; however, the amount and nature of such data remain critical considerations [44]. The ACWR theory also includes further complexities, such as coupled and uncoupled ACWR methods, in addition to the distinctions between RA and EWMA [45]. We found that consistent monitoring of recovery status from pre-race training through the tapering phase—particularly by maximizing the TQR*ewma* ratio of the recent 7 days to the preceding 21–28 days—can provide valuable insights to enhance race performance in various training environments.

We used the TQR score as a subjective measure, which inherently allowed for intra-individual variability that could affect inter-individual differences in outcomes. To address this, TQR scores were converted to z-scores for each participant during the observation period, and analyses were performed accordingly. The results closely matched those of the unconverted data, with the highest odds ratio in the TQR*ewma* ratio observed for the 7d:21d at 2.48. Despite documented individual differences in tolerance to TL and recovery from training-induced fatigue [31], it remains unclear whether the results of this study coincidentally reflected participant characteristics. However, longitudinal monitoring of TQR reveals that z-TQR scores fluctuate depending on data collection methods; therefore, using unconverted TQR scores in actual training contexts might be more practical. Regarding the effect on actual race time, the Kruskal–Wallis test indicated significant differences in TQR scores and all TQR*ewma* ratios. The ratios of race time to season-best were higher in the lowest quartile (Q1) and lower in the highest quartile (Q4), with the greatest difference observed in the TQR*ewma* ratio of 7d:21d at 0.89%. Similarly, z-TQR*ewma* showed a maximum difference of 0.87% for the same ratio, corresponding to 0.435 s for a swimmer with a season-best time of 50 s in the 100 m freestyle. A previous study has reported tapering-induced performance improvements ranging between 0.5% and 6.0% [46], placing our findings toward the lower end of this spectrum. However, insufficient evidence exists to explain this discrepancy. Race performance is strongly influenced by physical, physiological, and technical factors [47,48,49]. To elucidate the contribution of subjective recovery status to performance, further studies with designs controlling for these variables are necessary. This study targeted collegiate swimmers belonging to a single team. The training regimen over the two seasons was planned by specific coaches, resulting in similar macro- and mesocycles. Since TQR scores depend on TL, the participants’ fatigue-recovery cycles might also resemble each other. Additionally, the study focused on college swimmers from a single team, which resulted in a limited sample size. Therefore, reproducibility studies across different settings, coaches, and athletes are warranted.

TQR scores, primarily collected before training sessions, correlate with sRPE, an internal training load indicator [27,29]. Additionally, Osiecki et al. reported a correlation between TQR collected 24 h after exercise and blood CK levels [50]. Although TQR correlates with internal load and objective indicators, this study did not explore these relationships. Further investigation into these correlations and the optimal timing for TQR collection may be beneficial.

## 5. Conclusions

This study highlights the significance of using the TQR scale to monitor recovery status and its impact on race performance among collegiate swimmers. We found that a favorable recovery status on race day is associated with improved performance outcomes. Additionally, optimizing the TQR*ewma* ratio of the most recent 7 days compared to the preceding 21–28 days may further enhance race performance. As tapering is a crucial strategy following high-intensity training, our results underscore the necessity of continuously monitoring athletes’ recovery status to prevent deconditioning due to overload and improve race performance. Using TQR as a monitoring tool can contribute significantly to enhancing athletes’ performance in various training environments.

## Figures and Tables

**Figure 1 sports-13-00139-f001:**
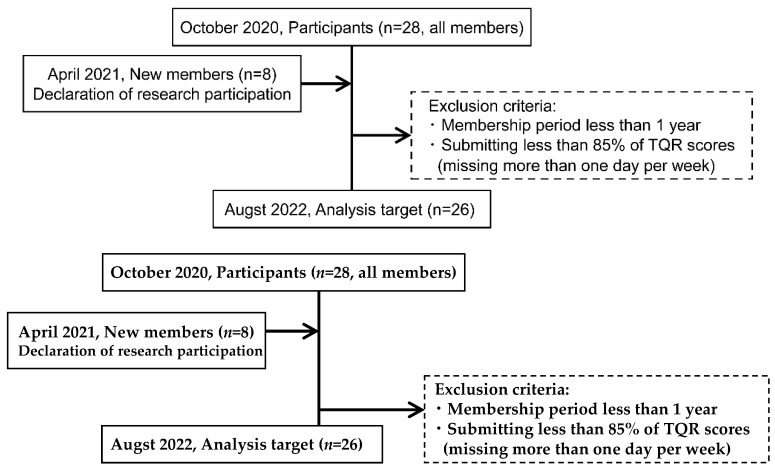
Flowchart for selecting analysis targets.

**Figure 2 sports-13-00139-f002:**
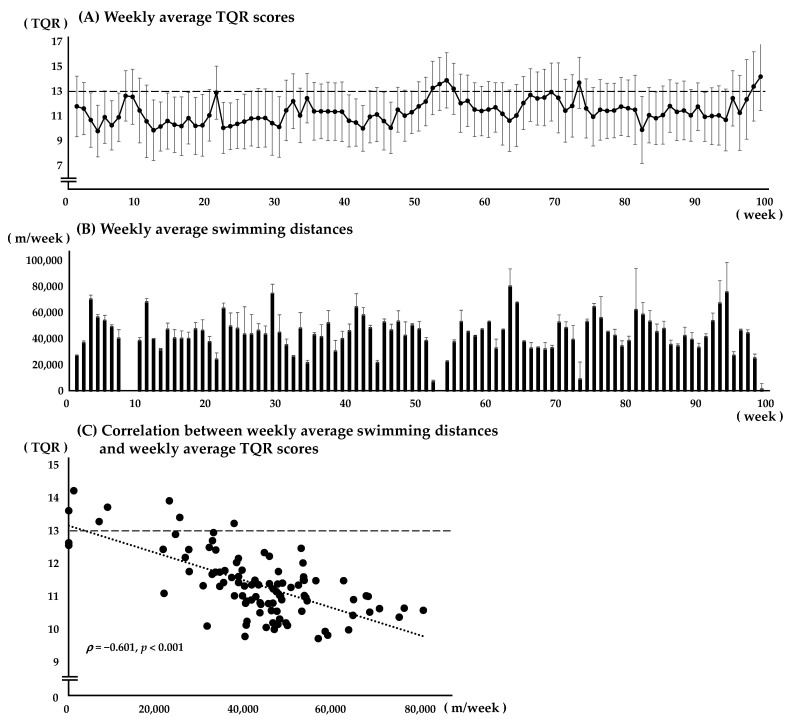
Weekly average TQR scores (**A**), swimming distances (**B**), and the correlation between these variables (**C**).

**Table 1 sports-13-00139-t001:** TQR score distribution for the observation period and race days.

TQR	6	7	8	9	10	11	12	13	14	15	16	17	18	19	20	Total
Total instances during observation period	359	322	868	1284	1251	1758	1689	1961	883	599	207	153	50	23	22	11,429
%	3.1	2.8	7.6	11.2	10.9	15.4	14.8	17.2	7.7	5.2	1.8	1.3	0.4	0.2	0.2	100.0
Instances during race days	11	11	31	39	54	63	94	115	58	44	13	11	3	2	1	550
%	2.0	2.0	5.6	7.1	9.8	11.5	17.1	20.9	10.5	8.0	2.4	2.0	0.5	0.4	0.2	100.0

Abbreviations: TQR: total quality recovery; *n*: number of observations; %: percentage of the total within the respective period.

**Table 2 sports-13-00139-t002:** Association between each TQR ratio and enhanced race performance in collegiate swimmers.

	Q1		Q2			Q3			Q4			*p* for Trend
	*n*		*n*	OR	95%CI	*n*	OR	95%CI	*n*	OR	95%CI	
					(Lower–Upper)			(Lower–Upper)			(Lower–Upper)	
TQR												
Raw	146	reference	157	1.46	(0.86–2.49)	115	2.22 ^†^	(1.28–3.87)	132	3.13 ^‡^	(1.84–5.30)	<0.001
TQR*ra*												
7d:14d	136	reference	139	1.40	(0.82–2.38)	137	1.79	(1.06–3.03)	138	1.66	(0.98–2.82)	0.037
7d:21d	140	reference	136	1.24	(0.74–2.10)	136	1.29	(0.76–2.17)	138	1.68	(1.01–2.80)	0.051
7d:28d	138	reference	138	1.04	(0.61–1.76)	137	1.38	(0.82–2.31)	137	1.72	(1.03–2.87)	0.019
14d:21d	137	reference	139	1.12	(0.67–1.88)	137	1.04	(0.62–1.74)	137	1.35	(0.81–2.24)	0.312
14d:28s	137	reference	139	1.21	(0.72–2.02)	137	1.11	(0.66–1.88)	137	1.54	(0.93–2.57)	0.133
21d:28d	136	reference	145	1.33	(0.81–2.18)	127	0.89	(0.52–1.50)	142	0.94	(0.56–1.57)	0.463
TQR*ewma*												
7d:14d	137	reference	138	1.07	(0.62–1.84)	140	1.58	(0.94–2.66)	135	2.15 ^†^	(1.28–3.60)	0.001
7d:21d	138	reference	138	1.30	(0.76–2.24)	136	1.48	(0.86–2.53)	138	2.62 ^‡^	(1.56–4.41)	<0.001
7d:28d	137	reference	140	1.37	(0.79–2.35)	135	1.71	(1.00–2.92)	138	2.48 ^‡^	(1.47–4.19)	<0.001
14d:21d	134	reference	142	1.29	(0.76–2.22)	138	1.49	(0.87–2.55)	136	2.30 ^†^	(1.36–3.88)	0.001
14d:28s	142	reference	133	1.13	(0.66–1.96)	138	1.80	(1.07–3.03)	137	2.13 ^†^	(1.27–3.57)	0.001
21d:28d	134	reference	142	1.20	(0.70–2.05)	142	1.47	(0.87–2.48)	132	2.19 ^†^	(1.30–3.69)	0.002

Data are presented as odd ratios (ORs) with 95% confidence intervals (CI: lower–upper) compared to a reference group. Q1, first quartile (lowest quartile); Q2, second quartile; Q3, third quartile; Q4, fourth quartile (highest quartile); OR, odds ratio; CI, confidence interval; TQR*ra*, rolling average of TQR; TQR*ewma*, exponentially weighted moving average of TQR. To assess the trend, the *p*-value was computed using the Cochran-Armitage test after performing the chi-square test. Statistical significance is indicated by ^†^ *p* < 0.005 and ^‡^ *p* < 0.001.

**Table 3 sports-13-00139-t003:** Relationship between each z-TQR ratio and enhanced race performance in collegiate swimmers.

	Q1		Q2			Q3			Q4			*p* for Trend
	*n*		*n*	OR	95% CI	*n*	OR	95% CI	*n*	OR	95% CI	
					(Lower Upper)			(Lower Upper)			(Lower Upper)	
z-TQR												
Raw	140	Reference	136	0.95	(0.53–1.70)	145	1.70	(1.00–2.91)	129	4.35 ^‡^	(2.56–7.41)	<0.001
z-TQR*ra*												
7d:14d	137	Reference	138	1.11	(0.65–1.88)	138	1.45	(0.86–2.43)	137	1.72	(1.03–2.87)	0.022
7d:21d	137	Reference	138	1.06	(0.63–1.80)	138	1.10	(0.65–1.86)	137	1.75	(1.05–2.90)	0.031
7d:28d	137	Reference	138	1.11	(0.65–1.88)	138	1.45	(0.86–2.43)	137	1.72	(1.03–2.87)	0.022
14d:21d	137	Reference	138	0.89	(0.53–1.49)	138	0.86	(0.51–1.44)	137	1.34	(0.81–2.20)	0.286
14d:28s	137	Reference	138	1.22	(0.73–2.03)	138	1.10	(0.65–1.85)	137	1.36	(0.81–2.26)	0.325
21d:28d	137	Reference	138	0.99	(0.60–1.63)	138	0.87	(0.52–1.44)	137	0.82	(0.49–1.36)	0.367
z-TQR*ewma*												
7d:14d	137	Reference	138	0.95	(0.55–1.65)	138	1.78	(1.06–2.99)	137	2.09 ^†^	(1.25–3.51)	0.001
7d:21d	137	Reference	138	1.16	(0.67–2.00)	138	1.69	(1.00–2.88)	137	2.48 ^‡^	(1.47–4.18)	<0.001
7d:28d	137	Reference	138	0.95	(0.55–1.65)	138	1.67	(0.99–2.81)	137	2.22 ^†^	(1.33–3.72)	<0.001
14d:21d	137	reference	138	1.03	(0.59–1.79)	138	1.85	(1.10–3.11)	137	2.18 ^†^	(1.30–3.66)	<0.001
14d:28s	137	reference	138	0.99	(0.57–1.71)	138	1.78	(1.06–2.99)	137	2.03	(1.21–3.41)	0.001
21d:28d	137	reference	138	0.99	(0.58–1.70)	138	1.31	(0.78–2.21)	137	2.06	(1.24–3.43)	0.002

Data are presented as odd ratios (ORs) with 95% confidence intervals (CI: lower–upper) compared to a reference group. Q1, first quartile (lowest quartile); Q2, second quartile; Q3, third quartile; Q4, fourth quartile (highest quartile); OR, odds ratio; CI, confidence interval; TQR*ra,* rolling average of TQR; TQR*ewma*, exponentially weighted moving average of TQR. To assess the trend, the *p*-value was calculated using the Cochran-Armitage test after performing the chi-square test. Statistical significance is indicated by ^†^ *p* < 0.005 and ^‡^
*p* < 0.001.

**Table 4 sports-13-00139-t004:** Race performance ratios compared to season-best times for each TQR ratio in collegiate swimmers.

	Q1		Q2		Q3		Q4		*p*
	Median	(IQR)	Median	(IQR)	Median	(IQR)	Median	(IQR)	
TQR									
race day	101.0	(100.1–101.9) ^b^	100.9	(99.9–101.8) ^b^	100.2	(99.7–101.4) ^a^	100.3	(99.6–101.0) ^a^	<0.001
TQR*ra*		–							
7d:14d	101.2	(100.0–102.3) ^b^	100.6	(99.8–101.3) ^a^	100.4	(99.8–101.3) ^a^	100.4	(99.6–101.4) ^a^	0.001
7d:21d	101.1	(100.0–102.1) ^b^	100.5	(99.8–101.4) ^ab^	100.6	(99.8–101.7) ^ab^	100.3	(99.6–101.2) ^a^	0.002
7d:28d	101.0	(100.0–102.0) ^b^	100.8	(99.8–101.7) ^ab^	100.5	(99.8–101.6) ^ab^	100.3	(99.6–101.2) ^a^	0.003
14d:21d	100.9	(99.9–101.9) ^b^	100.6	(99.8–101.7) ^ab^	100.7	(99.8–101.6) ^ab^	100.3	(99.8–101.2) ^a^	0.039
14d:28s	100.8	(99.9–101.8) ^n.s.^	100.5	(99.8–101.5)	100.8	(99.9–101.7)	100.3	(99.7–101.2)	0.032
21d:28d	100.6	(99.8–101.7) ^n.s.^	100.5	(99.6–101.3)	100.7	(100.0–101.6)	100.7	(99.9–101.7)	0.103
TQR*ewma*		–						–	
7d:14d	101.1	(100.0–101.9) ^c^	100.7	(99.8–101.7) ^bc^	100.5	(99.8–101.4) ^ab^	100.3	(99.6–101.1) ^a^	<0.001
7d:21d	101.1	(100.0–101.9) ^b^	100.6	(99.8–101.5) ^b^	100.7	(99.9–101.6) ^b^	100.2	(99.5–100.9) ^a^	0.001
7d:28d	101.1	(100.1–101.9) ^b^	100.6	(99.8–101.5) ^ab^	100.5	(99.8–101.7) ^ab^	100.3	(99.6–101.2) ^a^	<0.001
14d:21d	101.1	(100.0–102.0) ^b^	100.6	(99.8–101.5) ^ab^	100.6	(99.9–101.6) ^ab^	100.3	(99.6–101.2) ^a^	<0.001
14d:28s	101.1	(100.0–102.0) ^b^	100.6	(99.9–101.5) ^ab^	100.5	(99.8–101.5) ^ab^	100.3	(99.6–101.2) ^a^	0.001
21d:28d	100.9	(100.0–101.9) ^b^	100.7	(99.8–101.8) ^ab^	100.6	(99.8–101.5) ^ab^	100.3	(99.6–101.2) ^a^	0.002

Data are expressed as median (IQR). Data with different superscripts indicate significant differences at *p* < 0.05, as determined by the Mann–Whitney test. n.s.: not significant.

**Table 5 sports-13-00139-t005:** Race performance relative to season-best times across different z-TQR ratios in collegiate swimmers.

	Q1		Q2		Q3		Q4		*p*
	Median	(IQR)	Median	(IQR)	Median	(IQR)	Median	(IQR)	
z-TQR									
race day	101.0	(100.1–101.9) ^c^	100.9	(99.9–101.8) ^b^	100.2	(99.7–101.4) ^a^	100.3	(99.6–101) ^a^	<0.001
z-TQR*ra*									
7d:14d	101.1	(99.9–102.1) ^b^	100.6	(99.9–101.4) ^ab^	100.5	(99.8–101.3) ^a^	100.4	(99.6–101.4) ^a^	0.003
7d:21d	101.0	(99.9–101.9) ^b^	100.7	(99.9–101.4) ^ab^	100.7	(99.9–101.7) ^b^	100.3	(99.6–101.1) ^a^	0.001
7d:28d	101.1	(99.9–101.9) ^b^	100.7	(99.9–101.7) ^ab^	100.5	(99.8–101.5) ^ab^	100.3	(99.6–101.3) ^a^	0.003
14d:21d	100.8	(99.8–102.0) ^b^	100.8	(99.9–101.7) ^ab^	100.7	(99.9–101.6) ^ab^	100.3	(99.7–101.1) ^a^	0.027
14d:28s	100.8	(99.9–101.8) ^n.s.^	100.5	(99.8–101.7)	100.9	(99.8–101.8)	100.4	(99.8–101.2)	0.098
21d:28d	100.4	(99.8–101.5) ^n.s.^	100.6	(99.7–101.4)	100.7	(99.9–101.6)	100.7	(99.9–101.7)	0.477
z-TQR*ewma*		–							
7d:14d	101.1	(100.0–102.0) ^c^	100.8	(100.0–101.6) ^bc^	100.4	(99.8–101.3) ^ab^	100.3	(99.6–101.3) ^a^	<0.001
7d:21d	101.1	(100.0–101.9) ^c^	100.6	(99.9–101.6) ^bc^	100.5	(99.8–101.4) ^b^	100.2	(99.6–101.2) ^a^	<0.001
7d:28d	101.0	(100.0–101.9) ^b^	100.7	(100.0–101.9) ^b^	100.5	(99.8–101.4) ^ab^	100.3	(99.6–101.1) ^a^	<0.001
14d:21d	101.0	(100.0–101.9) ^c^	100.8	(100.0–101.9) ^bc^	100.5	(99.8–101.2) ^ab^	100.3	(99.6–101.3) ^a^	<0.001
14d:28s	101.0	(100.0–101.9) ^c^	100.7	(100.0–101.6) ^bc^	100.5	(99.8–101.4) ^ab^	100.3	(99.6–101.2) ^a^	0.001
21d:28d	100.9	(100.0–102.0) ^b^	100.7	(99.9–101.7) ^b^	100.6	(99.8–101.5) ^ab^	100.3	(99.6–101.2) ^a^	0.002

Data are expressed as Median (IQR). Data with different superscripts are significantly different at *p* < 0.05 (Mann–Whitney test). n.s.: not significant.

## Data Availability

The raw data supporting the conclusions of this article will be made available by the authors upon request.

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
