# Peer review of "Relationship Between the Total Quality Recovery Scale and Race Performance in Competitive College Swimmers over Two Seasons"

_sports, 2025, doi:10.3390/sports13050139_

Round 1
Reviewer 1 Report
Comments and Suggestions for Authors
Dear authors and editor,
The manuscript entitled “Relationship Between the Total Quality Recovery Scale and Race Performance in Competitive College Swimmers Over Two Seasons” has been reviewed. The paper presents important issues; however, adjustments are necessary for it to be published.
I kindly ask the authors to highlight all changes made to the manuscript after the first round of review in a different color or in yellow to make it easier for the reviewer to find the adjustments.
In general, the manuscript needs a general spelling review, since there are some places where the spelling punctuations need to be checked. For example, in the abstract, “;” always appears before a topic instead of “.”. At the end of the keywords, “)” appears after the period. Please read the manuscript carefully to adjust these details.
Abstract:
1- Make the objective clearer in the abstract, aligning it with what was presented at the end of the introduction. Both should be the same.
2- In line 16, is the age of the women, the men or all 22 participants? Please standardize. I believe that if you highlighted the women in the study, you should present the information for both separately.
3- Describe the type of study in methodology and some basic information about the participants, such as the category (youth, professional, amateur), average age of the athletes, time of practice and level (regional, national, international).
4- How were the athletes recruited? By lottery or by selection?
5- Explain how the periodization was carried out, what criteria did you use to monitor the athletes? Were biochemical analyses of lactate performed, for example?
6- What statistical program was used and the minimum significance level adopted?
7- What does the acronym show in line 20 mean: IQR?
8- In the description of the results, make the odds ratios clearer. What were they in relation to? In general terms, the description of the results needs to be better explained. It is difficult to understand the linearity of the work.
9- What does the acronym “Q4” shown in line 22 mean?
10- In conclusion, try to answer the objectives of the study more clearly. I thought it was too vague.
Keywords:
11- In keywords, please avoid using abbreviations and words that are repeated in the title.
Introduction:
12- In line 41, cite the reference number “Bosquet et al.’s”. Do the same in line 44 “Aouani et al.”. Check through the text for the need to adjust the references.
13- In the introduction, I missed mentioning the reference equipment and techniques to monitor athletes in relation to training periodization, such as the use of GPS, lactate, CK, CRP, VO2max, for example. Then, you introduce indirect methods, such as questionnaires on subjective perception of effort and recovery.
14- The last paragraph of the introduction is very confusing. Please adjust the objectives and make clearer what you intend to evaluate.
Materials and Methods:
15- The sample size calculation indicated the recruitment of 178 swimmers and you only managed to recruit 22. Please explain why you did not reach 12% of the recommended minimum?
16- Explain how the participants were recruited. I also did not find the clinical trial record. Did you register the study?
17- Explain in more detail the anthropometric assessments that were performed in the study. I did not find body mass index values ​​and other information that should appear in the first table of results. Present the information on body composition measured by the bioimpedance scale.
18- Please provide more details on how the athletes' training periodization was monitored. I did not find blood tests or GPS information, for example. How were the load adjustments made? Were physical tests performed to measure power, agility and VO2max?
19- I believe it is important for you to create a flowchart for the study, explaining how many people started and how many finished the research, presenting the reasons for the exclusions.
Results:
20- Create a table to present the sociodemographic, anthropometric and body composition characteristics of the athletes.
21- In figure 1, you could add figure 1C where you would present a correlation between the information in 1A and 1B. I believe that this would make it easier for the reader to follow the relationship between less training and greater recovery. This would also strengthen your use in the discussion.
22- Table 1 is disfigured, please adjust it to make it easier to understand. Also create a legend for abbreviations.
23- In tables 2 to 5, please present in the legend the meaning of Q1, Q2, Q3 and Q4. I believe that you could adjust the information in these tables so that the reader can check the information more clearly. The way the information is presented is confusing to follow.
Discussion:
24- I missed the authors presenting information in the literature about studies that performed similar monitoring with techniques using more objective standards such as CK and VO2max. Please include these studies, showing that the protocol used by you is representative of the more objective standards for assessing recovery.
25- Present the limitations of the study at the end of the discussion. One of them is the low number of participants, proven by the sample size calculation.
Conclusions:
26- In conclusion, please try to focus on the objectives of the study.
Author Response
Response to Reviewer’s Comments
Referee comment 1
We would like to sincerely thank you for your careful and thorough reading of the manuscript, as well as for your thoughtful comments and constructive suggestions, which have helped us improve the quality of the article. All requested changes have been marked in yellow. We greatly appreciate your advice and the prompt peer review. Please find our point-by-point response below. Thank you.
In general, the manuscript needs a general spelling review, since there are some places where the spelling punctuations need to be checked. For example, in the abstract, “;” always appears before a topic instead of “.”. At the end of the keywords, “)” appears after the period. Please read the manuscript carefully to adjust these details.
Reply:
Thank you for pointing this out. We apologize for these mistakes. We have corrected the issues you mentioned. Thank you again for your valuable comment!
1- Make the objective clearer in the abstract, aligning it with what was presented at the end of the introduction. Both should be the same.
Reply:
Thank you for pointing this out. We have added the content from the abstract to the end of the Introduction, as follows:
In this study, we attempted to investigate the impact of recovery status on race performance. We longitudinally monitored the TQR scale in competitive swimmers throughout the season and examined their pre-race recovery status using TQR-related variables.
2- In line 16, is the age of the women, the men or all 22 participants? Please standardize. I believe that if you highlighted the women in the study, you should present the information for both separately.
Reply:
Thank you for pointing this out. The previous wording was confusing; however, the reported age represents the average of the 22 participants. Therefore, we have revised the content as follows.
Total quality recovery (TQR) scale scores were monitored over two seasons in 22 college competitive swimmers (age: 19.7 ± 1.8 years), including six females.(Lines 16–17)
3- Describe the type of study in methodology and some basic information about the participants, such as the category (youth, professional, amateur), average age of the athletes, time of practice and level (regional, national, international).
Reply:
Thank you for your comment. Basic information about the participants, including average age, height, weight, and competition history, is provided in Lines 110–113. Regarding competition level, since levels vary by country, we have cited Ruiz-Navarro et al. [33], which categorizes competition levels based on FINA points. Because it is difficult to measure the exact daily practice time—due to travel time between the gym and the pool, as well as rest periods—general training hours are provided in Lines 124–126.
4- How were the athletes recruited? By lottery or by selection?
Reply:
Thank you for pointing this out. During a team meeting, we explained the study objectives to all swimmers and distributed consent and withdrawal forms, after which we obtained their voluntary participation. Therefore, the following sentence was added to the abstract (Line 17):
They participated voluntarily in the study.
5- Explain how the periodization was carried out, what criteria did you use to monitor the athletes? Were biochemical analyses of lactate performed, for example?
Reply:
Thank you for your comment. Periodization during the study was managed by the team coaches, and no individual participant monitoring was performed.
6- What statistical program was used and the minimum significance level adopted?
Reply:
Thank you for your comment. The statistical software and significance levels used in this study are described in Section 2, “Materials and Methods “(Lines 186–187).
7- What does the acronym show in line 20 mean: IQR?
Reply:
Thank you for pointing this out. We apologize for these errors. IQR is an abbreviation for Interquartile Range; therefore, we have corrected it as follows (Line 21):
The median TQR was 11 (interquartile range: 10–13).
8- In the description of the results, make the odds ratios clearer. What were they in relation to? In general terms, the description of the results needs to be better explained. It is difficult to understand the linearity of the work.
Reply:
Thank you for pointing this out. The odds ratios were added as follows (Lines 22–25):
The highest odds ratios (ORs) for performance improvement in the highest quartile (Q4) of TQR and z-TQR were 3.13 (p < 0.001) and 4.35 (p < 0.001), respectively. Significant ratios for TQRewma were observed for 7d:21d (OR: 2.62, p < 0.001) and 7d:28d (OR: 2.48, p < 0.001) comparisons.
Additionally, TQRra and TQRewma in Lines 17, 18, and 24 were not written in italics. We discovered this error after receiving your feedback and have now made the necessary corrections. We apologize for the inconvenience.
9- What does the acronym “Q4” shown in line 22 mean?
Reply:
Thank you for pointing this out. Q4 represents the highest quartile, as mentioned in my response to comment number 8.
The highest odds ratios (ORs) for performance improvement in the highest quartile (Q4) of TQR and z-TQR were 3.13 (p < 0.001) and 4.35 (p < 0.001), respectively. Significant ratios for TQRewma were found for 7d:21d (OR: 2.62, p < 0.001) and 7d:28d (OR: 2.48, p < 0.001) comparisons.
10- In conclusion, try to answer the objectives of the study more clearly. I thought it was too vague.
Reply:
Thank you for pointing this out. The following corrections have been made (Lines 26–30):
Better recovery status on race day has been associated with improved swimming performance. Additionally, optimizing the TQRewma ratio of the most recent 7 days compared to the preceding 21 to 28 days may further enhance race performance. This highlights the need to monitor an athlete’s recovery over several weeks as an important pre-race strategy.
11- In keywords, please avoid using abbreviations and words that are repeated in the title.
Reply:
Thank you for pointing this out. The following corrections have been made (Lines 31–32):
swimming performance; recovery state; total quality recovery; daily monitoring; tapering strategy; subjective recovery assessment; training load management
12- In line 41, cite the reference number “Bosquet et al.’s”. Do the same in line 44 “Aouani et al.”. Check through the text for the need to adjust the references.
Reply:
Thank you for pointing this out. We apologize for these errors. The following corrections have been made (Lines 46–51):
Bosquet et al.’s meta-analysis revealed that effective tapering strategies often involve a gradual reduction in training volume by 41–60% over 8–14 days while maintaining training intensity and frequency [11]. They further emphasized the importance of maintaining training intensity and frequency during tapering. Aouani et al. found that a 2-week tapering period improved performance in the 50-m front crawl test in competitive swimmers [12].
13- In the introduction, I missed mentioning the reference equipment and techniques to monitor athletes in relation to training periodization, such as the use of GPS, lactate, CK, CRP, VO2max, for example. Then, you introduce indirect methods, such as questionnaires on subjective perception of effort and recovery.
Reply:
Thank you for your comment. The purpose of this study is to propose a training plan for the adjustment period aimed at achieving optimal race performance using subjective indicators. Therefore, as you pointed out, there is insufficient mention of physiological indicators such as GPS, lactate levels, CK, CRP, and VO2max. Although we mentioned the relationship with physiological indicators in the discussion (Lines 343–345), these indicators were not evaluated in this study and are mentioned as limitations. Furthermore, to explain the usefulness of subjective scales, we cite a report by Saw et al. in the Introduction (Lines 73–76).
14- The last paragraph of the introduction is very confusing. Please adjust the objectives and make clearer what you intend to evaluate.
Reply:
Thank you for pointing this out. We understand that this comment is related to comment number 1. Therefore, we have amended it as follows (Lines 89–91):
In this study, we investigated the impact of recovery status on race performance. We monitored the TQR score longitudinally in competitive swimmers throughout the season and examined their pre-race recovery status using TQR-related variables.
15- The sample size calculation indicated the recruitment of 178 swimmers and you only managed to recruit 22. Please explain why you did not reach 12% of the recommended minimum?
Reply:
Thank you for pointing this out. As you pointed out, the 22 recruits represent the sample size, which is less than the required sample size of 178. For this reason, we have removed the statement regarding the sample size estimate (Lines 103–106) and listed the following as limitations (Lines 338–339):
Additionally, the study focused on college swimmers from a single team, which resulted in a limited sample size.
16- Explain how the participants were recruited. I also did not find the clinical trial record. Did you register the study?
Reply:
Thank you for pointing it out. We added the following (Lines 99–100):
Participants were recruited during team meetings, and all swimmers volunteered.
Unfortunately, we did not register this study in any clinical trial registry.
17- Explain in more detail the anthropometric assessments that were performed in the study. I did not find body mass index values ​​and other information that should appear in the first table of results. Present the information on body composition measured by the bioimpedance scale.
Reply:
Thank you for pointing it out. We added the following (Lines 113–114):
the average lean body mass was 53.7 ± 8.4 kg, and the average body fat mass was 10.2 ± 2.6 kg.
18- Please provide more details on how the athletes' training periodization was monitored. I did not find blood tests or GPS information, for example. How were the load adjustments made? Were physical tests performed to measure power, agility and VO2max?
Reply:
Thank you for your comment. As stated in comment 13, blood tests and GPS data were not collected in this study. We responded to comment 13 as follows:
Reply to comment 13:
Thank you for your comment. The purpose of this study is to propose a training plan for the adjustment period to achieve optimal performance in the race using subjective indicators. Therefore, as you pointed out, there is insufficient mention of physiological indicators such as GPS, lactate levels, CK, CRP, and VO2max. Although we discussed the relationship with physiological indicators in the discussion (Lines 343–345), these indicators were not evaluated in this study and are acknowledged as limitations. Furthermore, to explain the usefulness of subjective scales, we cite a report by Saw et al. in the Introduction (Lines 73–76).
19- I believe it is important for you to create a flowchart for the study, explaining how many people started and how many finished the research, presenting the reasons for the exclusions.
Reply:
Thank you for your comment. The exclusion criteria for participants included in the analysis are listed in Lines 100–102. Based on your suggestions, we have created a flowchart (Figure 1).
Accordingly, “(Figure 1)” has been added in Line 102.
The study focused on 22 swimmers who had been members of the team for at least one season during the monitoring period and who had a TQR score submission rate above 85% (missing no more than 1 day per week) (Figure 1).
20- Create a table to present the sociodemographic, anthropometric and body composition characteristics of the athletes.
Reply:
Thank you for your comment. As stated in the response to comment 17, we added information about the physical characteristics in Section 2.1 “Participants.” Therefore, we have not created a table.
21- In figure 1, you could add figure 1C where you would present a correlation between the information in 1A and 1B. I believe that this would make it easier for the reader to follow the relationship between less training and greater recovery. This would also strengthen your use in the discussion.
Reply:
Thank you for pointing this out. Following your suggestion, we created a correlation chart showing the relationship between average weekly swimming distance and average TQR scores and added it to Figure 2.
22- Table 1 is disfigured, please adjust it to make it easier to understand. Also create a legend for abbreviations.
Reply:
Thank you for pointing this out. Following your suggestion, we added a legend for abbreviations to Table 1.
23- In tables 2 to 5, please present in the legend the meaning of Q1, Q2, Q3 and Q4. I believe that you could adjust the information in these tables so that the reader can check the information more clearly. The way the information is presented is confusing to follow.
Reply:
Thank you for pointing this out. As instructed, we have added explanations for Q1, Q2, Q3, and Q4 to the legends in Tables 2 and 3.
Data are presented as odds ratios (ORs) with 95% confidence intervals (CI: lower–upper) compared to a reference group. Q1, first quartile (lowest quartile); Q2, second quartile; Q3, third quartile; Q4, fourth quartile (highest quartile); OR, odds ratio; CI, confidence interval; TQRra, rolling average of TQR; TQRewma, exponentially weighted moving average of TQR. To assess the trend, the p-value was computed using the Cochran-Armitage test after performing the chi-square test. Statistical significance is indicated by †p < 0.005 and ‡p < 0.001.
24- I missed the authors presenting information in the literature about studies that performed similar monitoring with techniques using more objective standards such as CK and VO2max. Please include these studies, showing that the protocol used by you is representative of the more objective standards for assessing recovery.
Reply:
Thank you for your comments. As mentioned in comments 13 and 18, we were unable to monitor objective indicators in this study. The usefulness of subjective indicators is described in the Introduction, citing an article by Saw et al. (Lines 73–76). To avoid duplication, we did not mention it in the discussion.
In the field of sports, monitoring objective indicators daily is often unrealistic. In this study, we examined the possibility of using a subjective scale (TQR). We believe that the relationship between subjective and objective scales needs to be examined in the future and have stated this as a limitation (Lines 343–345).
Reply to comment 13:
Thank you for your comment. This study aims to propose a training plan for the adjustment period to achieve optimal race performance using subjective indicators. Therefore, as you pointed out, there is insufficient reference to physiological indicators such as GPS, lactate levels, CK, CRP, and VO2max. Although we discussed the relationship with physiological indicators in the discussion (Lines 343–345), these indicators were not evaluated in this study and are mentioned as limitations. Furthermore, to explain the usefulness of subjective scales, we cite a report by Saw et al. in the introduction (Lines 73–76).
25- Present the limitations of the study at the end of the discussion. One of them is the low number of participants, proven by the sample size calculation.
Reply:
Thank you for your comment. As pointed out in comment 15, we have added the following:
Additionally, the study focused on college swimmers from a single team, which resulted in a limited sample size.
Reply to comment 15:
Thank you for pointing this out. As you pointed out, the 22 recruits represent the sample size, which is smaller than the required sample size of 178. For this reason, we have removed the statement regarding the sample size estimate (Lines 103–106) and listed the following as limitations (Lines 338–339):
Additionally, the study focused on college swimmers from a single team, which resulted in a limited sample size.
26- In conclusion, please try to focus on the objectives of the study.
Reply:
Thank you for pointing it out. Following your suggestion, we have fixed it as follows:(Lines 352–355.):
As tapering is a crucial strategy following high-intensity training, our results underscore the necessity of continuously monitoring athletes’ recovery status to prevent deconditioning caused by overload and improve race performance.
Finally, we would like to thank you again for carefully reading this manuscript and for your precise and valuable feedback! We appreciate your comments, which helped us improve the manuscript. Thank you!
Reviewer 2 Report
Comments and Suggestions for Authors
Thank you for the opportunity to review your manuscript examining the impact of recovery on race performance in collegiate swimmers. Your findings of athletes who had better recovery, via the TQR scale, had improved race performance is significant. While most athletes would antidotally tell you they perform better when well recovered, this is an important addition to the literature. I do have a couple comments that I feel would improve your mansucript.
Page 4, Lines 148-149: Please explain in detail how the TQR ratios were converted to z-scores.
Page 14, Line 311-312: You state "Previous studies...." but only reference 1 study. Include additional references or rephrase to be singular.
Author Response
Response to Reviewer’s Comments
Referee comment 2
We would like to thank you very much for your careful and thorough reading of this manuscript, as well as for the thoughtful comments and constructive suggestions, which helped us improve the quality of this article. All requested changes have been clearly marked in yellow. We greatly appreciate your advice and the fast peer review. Please find our point-by-point response below. Thank you
Page 4, Lines 148-149: Please explain in detail how the TQR ratios were converted to z-scores.
Reply:
Thank you for pointing it out. Following your suggestion, I added the following (Lines 161–165):
The z-score is a standardized score that follows a normal distribution with a mean of 0 and a standard deviation of 1. Each individual’s z-score was calculated by dividing the difference between their daily TQR score and their average TQR score during the observation period by the standard deviation.
Page 14, Line 311-312: You state “Previous studies....” but only reference 1 study. Include additional references or rephrase to be singular.
Reply:
Thank you for pointing it out. We apologize for these mistakes. It has been corrected as follows (Line 329):
Previous study has reported tapering-induced performance improvements ranging between 0.5% and 6.0% [46],
Finally, we would like to thank you again for the careful reading of this manuscript and for your precise and valuable feedback! We appreciate your comments, which helped us improve the manuscript. Thank you!
Reviewer 3 Report
Comments and Suggestions for Authors
The article is structured in accordance with the stages of scientific research. The bibliography is well chosen and supports the research conducted.
Author Response
Response to Reviewer’s Comments
Referee comment 3
The article is structured in accordance with the stages of scientific research. The bibliography is well chosen and supports the research conducted.
We would like to thank you very much for your careful and thorough reading of this manuscript. Thank you.
Round 2
Reviewer 1 Report
Comments and Suggestions for Authors
Dear authors,
Thank you for sending the manuscript with the requested adjustments and explanations for the issues addressed. After reviewing the changes in the manuscript, my opinion is that the work should be approved for publication.